# Synergistic Effect of Conditioned Medium from Amniotic Membrane Mesenchymal Stromal Cells Combined with Paclitaxel on Ovarian Cancer Cell Viability and Migration in 2D and 3D In Vitro Models

**DOI:** 10.3390/pharmaceutics17040420

**Published:** 2025-03-26

**Authors:** Paola Chiodelli, Patrizia Bonassi Signoroni, Elisa Scalvini, Serafina Farigu, Elisabetta Giuzzi, Alice Paini, Andrea Papait, Francesca Romana Stefani, Antonietta Rosa Silini, Ornella Parolini

**Affiliations:** 1Department of Life Science and Public Health, Università Cattolica del Sacro Cuore, 00168 Rome, Italy; andrea.papait@unicatt.it (A.P.); ornella.parolini@unicatt.it (O.P.); 2Centro di Ricerca E. Menni, Fondazione Poliambulanza Istituto Ospedaliero, 25124 Brescia, Italy; patrizia.bonassi@poliambulanza.it (P.B.S.); elisa.scalvini@poliambulanza.it (E.S.); serafina.farigu@poliambulanza.it (S.F.); elisabetta.giuzzi@poliambulanza.it (E.G.); alice.paini@poliambulanza.it (A.P.); francesca.stefani@poliambulanza.it (F.R.S.); antonietta.silini@poliambulanza.it (A.R.S.); 3Fondazione Policlinico Universitario “Agostino Gemelli” IRCCS, 00168 Rome, Italy; 4Fondazione IRCCS Casa Sollievo della Sofferenza, Viale Cappuccini 1, San Giovanni Rotondo, 71013 Foggia, Italy

**Keywords:** human amniotic mesenchymal stromal cell, amniotic membrane, ovarian cancer, 3D model, paclitaxel

## Abstract

**Background:** Ovarian cancer accounts for more deaths than any other cancer of the female reproductive system. Despite standard care, recurrence due to tumor spread and chemoresistance is common, highlighting the need for novel therapies. Mesenchymal stromal cells from the human amniotic membrane (hAMSC) and the intact amniotic membrane (hAM) are promising due to their secretion of tumor-modulating bioactive factors, accessibility from biological waste, and ethical favorability. Furthermore, unlike isolated cells, hAM provides an easier, clinically translatable product. We previously demonstrated that hAMSC can inhibit tumor cell proliferation, both in contact and transwell settings, suggesting that hAMSC secrete bioactive factors able to target tumor cells. This study evaluates the anti-tumor effects of bioactive factors from hAMSC and hAM conditioned medium (CM) on ovarian cancer cells in 2D and 3D models, alone or with paclitaxel. **Methods:** The impact of CM, alone or with paclitaxel, was tested on ovarian cancer cell proliferation, migration, invasion, and on angiogenesis. **Results:** hAMSC-CM and hAM-CM inhibited the proliferation and migration in 2D cultures and reduced spheroid growth and invasion in 3D models. Combining CM with paclitaxel enhanced anti-tumor effects in both settings. **Conclusions:** hAMSC-CM and hAM-CM show therapeutic potential against ovarian cancer, with synergistic benefits when combined with paclitaxel.

## 1. Introduction

Ovarian cancer is among the most lethal gynecological malignancies, characterized by late diagnosis, high recurrence rates, and resistance to conventional therapies [1]. Despite advancements in chemotherapy, including the widespread use of paclitaxel and platinum-based agents, the five-year survival rate remains dismally low (about 31%) for advanced-stage disease [2,3,4], underscoring the urgent need for novel therapeutic strategies.

Recent studies have highlighted the dual nature of mesenchymal stromal cells (MSC) in cancer biology; while they can promote tumor growth and metastasis under certain conditions, their conditioned media (CM) demonstrate notable anti-tumor effects [5,6,7,8]. MSC from different sources are known for their unique ability to secrete a diverse array of bioactive molecules, including cytokines, growth factors, and extracellular vesicles, which collectively form a complex secretome. This secretome plays a pivotal role in mediating the paracrine effects. The therapeutic potential of the MSC secretome, or CM lies in its ability to modulate key biological processes such as proliferation, apoptosis, and migration affecting different cellular pathways including IL-6/JAK2/STAT3, cyclins, FAK/PI3K/Akt/mTOR [9,10,11].

The human amniotic membrane is a highly favorable source of mesenchymal stromal cells (hAMSC) due to its origin from placental tissue after childbirth, making it an ethically acceptable and non-controversial option. Furthermore, it is obtained from biological waste—the placenta—following routine childbirth, ensuring no harm or risk to the mother or baby. hAMSC meet the established criteria for MSC from all tissues [12], as well as the specific standards for fetal membrane MSC defined during the First International Workshop on Placenta-Derived Stem Cells in 2008 [13]. On the other hand, the intact amniotic membrane (hAM) comprises two cell populations, hAMSC and amniotic epithelial cells (hAEC), both of which exhibit anti-tumor properties [14,15,16]; hAM is easier to use, as it bypasses the additional step of MSC isolation.

In our previous research, we showed that hAMSC can effectively inhibit tumor cell proliferation in a paracrine manner [17], and that hAM-CM reduces the migration of bladder urothelial cancer cell lines [18]. Notably, both hAMSC-CM and hAM-CM are rich in cytokines, growth factors, and extracellular vesicles (EVs) [19,20].

The aim of this study is to evaluate the anticancer potential of hAMSC-CM and hAM-CM on ovarian cancer cell lines (HEY, OV-90, and SKOV3) using the classical 2D monolayer and the more advanced 3D spheroid culture system, a physiologically relevant environment closely mimicking in vivo tumor architecture and behavior. Given that 3D cell cultures more accurately mimic in vivo tumor conditions, including increased resistance to chemotherapeutic agents such as paclitaxel, we hypothesize that combining paclitaxel with CM might overcome this resistance in 3D ovarian cancer models.

## 2. Materials and Methods

### 2.1. Cell Cultures

HEY and OV-90 human ovarian cancer cell lines were kindly provided by Daniela Gallo (Fondazione Policlinico Universitario “Agostino Gemelli” IRCCS, Rome, Italy). SKOV3 cell line was purchased by ATCC. HEY cells derive from a human ovarian cancer xenograft originally grown from a peritoneal deposit of a patient with moderately differentiated papillary cystadenocarcinoma [21]. OV-90 cells were originally isolated from malignant ascites from patients with ovarian adenocarcinoma and harbor p53 mutations that exhibit genomic features similar to high-grade serous ovarian carcinoma. SKOV3 cells derive from the ascites of a patient with ovarian adenocarcinoma and lack the expression of p53 protein. HEY cells were cultured in RPMI 1640 (Euroclone, Pero, Itlay; ECB9006) supplemented with 10% fetal bovine serum (FBS) (Euroclone; ECS50000LH) and 1% non-essential amino acids (Thermo Fisher, Waltham, MA, USA; 11140-035). OV-90 cells were cultured in a complex medium composed by MCDB (Sigma Aldrich, St. Louis, MO, USA; M-6770) plus M199 (Merk; Whitehouse Station, NJ, USA; M4530) at a 1:1 ratio, and 0.5% MEM (Sigma Aldrich; 56416C-1L), supplemented with 15% FBS. SKOV3 cells were cultured in Mcoy’s medium (Euroclone, ECM0210L) supplemented with 10% FBS. All culture media were supplemented with 2 mM L-glutamine (Euroclone; ECB3000D) and 1% penicillin/streptomycin (herein referred to P/S, all from Euroclone; ECB30010D).

Umbilical cords were obtained following the guidelines outlined by the Brescia Provincial Ethics Committee (number NP 2243, 19/01/2016). Human umbilical vein endothelial cells (HUVEC) were isolated from the umbilical cord vein following cannulation, followed by PBS washes to remove blood cells. The vein was then incubated for 1 h at 37 °C in DMEM supplemented with P/S, 0.2% collagenase I (Thermo Fisher; 17018029), and 0.01 mg/mL DNase (Merck, Darmstadt, Germany; 11284932001). After incubation, the detached cells were collected using PBS (Euroclone; ECB4004) with P/S, centrifuged at 300× *g* for 10 min without braking, resuspended in medium, and seeded into a flask pre-coated with 1.5% porcine gelatin (Merck; G2500). HUVEC at passages I–V were grown in EBM-2 Basal Medium (Lonza, Basilea, Swizerland; CC-3156) supplemented with EGM-2 SingleQuots Supplements (Lonza; CC-4176).

### 2.2. Human Amniotic Membrane (hAM) Fragment Preparation

The study adhered to the principles of the Declaration of Helsinki, and informed consent was obtained following the guidelines outlined by the Brescia Provincial Ethics Committee (number NP 2243, 19 January 2016).

For each placenta, the amniotic membrane was manually separated from the chorion and cut into a 50 cm^2^ fragments for CM preparation from hAM, or into 9 cm^2^ for the isolation of hAMSCs. The fragments were decontaminated by placing them in a physiological solution + 0.25% povidone-iodine for 1–2 s, then removed and incubated in PBS + P/S + amphotericin B (Euroclone; ECM0009D) + cefamezin (Teva Italia, Milan, Italy) for 3 min. The fragments were then washed in PBS containing 1% P/S.

### 2.3. Isolation and Culture of Human Amniotic Mesenchymal Stromal Cells (hAMSC)

Cells were isolated and phenotypically characterized as previously described [22]. hAM fragments were digested at 37 °C for 9 min with 2.5 U/mL dispase (VWR, Radnor, PA, USA; 734-1312) and then transferred to RPMI 1640 medium supplemented with 10% heat-inactivated FBS, 1% P/S, and 1% L-glutamine. Afterward, the fragments were treated with 0.9 mg/mL collagenase (Merck; 10103586001) and 0.01 mg/mL DNase I (Merck; 11284932001) for approximately 2.5–3 h at 37 °C. The resulting cell suspensions were centrifuged at 150 g, and the supernatant was filtered through a 100-μm cell strainer (BD Falcon, Bedford, MA, USA), and the cells were collected by centrifugation. Freshly isolated cells were expanded until passage 2 (p2) by plating at a density of 10,000 cells/cm^2^ in Chang medium D (Irvine Scientific, Santa Ana, CA, USA; 12401340) supplemented with 2 mM L-glutamine and 1% P/S at 37 °C in the incubator at 5% CO_2_ and phenotypically characterized.

### 2.4. Preparation of Conditioned Medium (CM)

From hAM: two fragments of hAM were placed in 50 mL conical tubes with filter caps (Greiner Bio-One, Kremsmünster, Austria; 227245) containing 20 mL of DMEM/F12 (Thermo Fisher Scientific; 31330038). The fragments were left for 5 days under gentle rotation at 37 °C and 5% CO₂. After incubation, the medium was centrifuged at 300× *g*, filtered through a 0.2-μm sterile filter (Sartorius Stedim, Florence, Italy: 16532), and stored at −80 °C until use.

From hAMSC P2: hAMSC were cultured for 5 days in 24-well plates (Euroclone; ET3024) at a density of 5 × 10⁵ cells/well in 0.5 mL of DMEM/F12 supplemented with 2 mM L-glutamine and 1% P/S as previously described [23]. At the end of incubation, the CM was collected, centrifuged at 300× *g*, filtered through a 0.2-μm sterile filter (Sartorius Stedim), and stored at −80 °C until use.

Pooling of CM for experiments: Each experiment was performed using CM pooled from at least three different hAMSC/hAM donors. The CM were used at varying percentages in DMEM/F12 as required by the experimental protocol. For each experiment, DMEM/F12 served as the negative control. FBS was added to both CM, and the negative control at the concentration specified for each assay. In all assays, CM and the negative control were added simultaneously, using the same volume and experimental procedures.

### 2.5. Determination of Cellular Viability (MTT-Assay and CyQUANT-Assay)

Cells were plated in 96-well plates at a density of 15,000 cells/cm^2^ for HEY and SKOV3 or 30,000 cells/cm^2^ for OV-90. They were then treated with increasing concentrations of different CM (12.5%, 25%, 50%, 75%, 100%) in DMEM/F12 containing 1% FBS for 24, 48, 72, and 96 h.

For the matrix assay, cells were treated for 48 h with increasing concentrations of CM (12.5%, 25%, 50%, 75%, 100%) in combination with increasing concentrations of paclitaxel (Vinci-Biochem, Vinci, Italy; Cod. AG-CN2-0045-M001) (0.6, 1.8, 5.5, 16.6, 50 nM). At the end of incubation, cell viability was assessed using either the MTT or CyQUANT assay.

MTT Assay: MTT (Merck; M2128) was added to each well at a final concentration of 0.5 mg/mL, and the cells were incubated at 37 °C for 3 h. After incubation, 100 µL of lysis buffer [20% SDS (Merck; 1.13760.0100) in a 50% H₂O/50% DMF (Thermo Fisher Scientific, 423640010) solution, pH 4.7] was added overnight at 37 °C. The absorbance was measured at 550 nm using a Victor™ X4 plate reader (PerkinElmer, Waltham, MA, USA). Since MTT occurs only in metabolically active cells, the level of activity was considered as an indirect measure of the viability of the cells, without a count of live and death cells.

CyQuant Assay: The CyQUANT™ NF Cell Proliferation Assay (Thermo Fisher Scientific; C35006) was performed according to the manufacturer’s instructions. Briefly, the medium was removed from each well, and 100 µL of CyQuant reagent was added. The cells were incubated at 37 °C for 1 h. Fluorescence was then measured using a Victor™ X4 plate reader at 485 nm.

### 2.6. Colony Formation

Cells were seeded into 12-well plates (Euroclone; ET3012) at the following densities: HEY (250 cells/well), SKOV3 (500 cells/well), and OV-90 (800 cells/well). Cells were allowed to attach overnight. The next day, cells were treated with 50%, 75%, or 100% of the different CM in 1% FBS and cultured for 10–14 days to allow colony formation. After the incubation period, debris was washed twice with PBS, and cells were fixed with cold methanol for 10 min. They were then stained with crystal violet (Merck; V5265). After staining, the cells were solubilized with 400 µL of 30% acetic acid (Carlo Erba, Cornaredo, MI, Italy; 524521). A 100 µL aliquot was transferred to a 96-well plate, and absorbance was measured at 550 nm using a Victor™ X4 plate reader.

### 2.7. Apoptosis Assay

Cell apoptosis was evaluated using flow cytometry with the Annexin V-fluorescein isothiocyanate (FITC)/Propidium iodide (PI) kit (BD Pharmingen™ FITC Annexin V kit, Cat. No. 556420). Cells were treated with different concentrations of CM (50%, 75%, or 100%) in DMEM/F12 containing 1% FBS. After treatment, the cells were resuspended in buffer and incubated with Annexin V at room temperature for 20 min. Subsequently, PI was added and the cells were further incubated for an additional 10 min at room temperature. Samples were acquired on a BD FACS Symphony A3 flow cytometer (BD Biosciences, Franklin Lakes, NJ, USA), and the data were analyzed using FlowJo 10.8 software (BD Biosciences).

### 2.8. Wound Healing Assay

Cells were seeded in silicone inserts (ibidi GmbH, Gräfelfing, Germany; 81176) in 24-well plates. The following cell densities were used: HEY at 100,000 cells/cm^2^, SKOV3 at 85,000 cells/cm^2^, and OV-90 at 150,000 cells/cm^2^. The day after seeding, the inserts were removed, and the cells were treated with 50%, 75%, or 100% of the different CM in 1% FBS. Microphotographs were taken at Day 0, Day 1, and Day 2 using an Olympus IX50 microscope equipped with an OPTIKA camera (OPTIKA, Ponteranica, Italy; Model 4083.13), using a 4× magnification. The extent of wound repair was evaluated by measuring the wound area at Day 1 and Day 2 relative to Day 0 (T0) using computerized image analysis with ImageJ software (http://rsb.info.nih.gov/ij/).

### 2.9. Single-Cell Migration

Cell motility was assessed by time-lapse videomicroscopy. HEY cells were seeded at a density of 3000 cells/cm^2^, and SKOV3 and OV-90 cells were seeded at 5000 cells/cm^2^ in 12-well plates. After 18 h, the cells were treated with 75% of the different CM in 1% FBS. A constant temperature of 37 °C and pCO₂ of 5% were maintained throughout the experimental period. Cells were observed under Mica Widefield Live Cell (Leica Microsystems, Wetzlar, Germany), and images (1 frame every 30 min) were digitally recorded for 1380 min. Single-cell migration was analyzed using AIVIA software version 12.1 (Leica Microsystems) through cell tracking recipe. Path lengths (in µm) of cells were recovered for at least 10 frames and were used for analysis. Path Length is the distance traveled by the object along its trajectory over the entire duration of the track and is calculated as the sum of the Euclidean distances between the object’s positions at sequential time points.

### 2.10. Transwell Migration Assay

Cells were seeded in 6-well plates at a density of 50,000 cells/cm^2^ and treated with 50%, 75%, or 100% of the different CM plus 1% FBS for 24 h. After treatment, the cells were detached and seeded in transwell inserts (Corning, Glendale, AZ, USA; CL-β422-48EA) at the following densities: 100,000 cells/insert for HEY and OV-90, 40,000 cells/insert for SKOV3, in a volume of 150 µL of serum-free DMEM/F12. The inserts were then placed into 24-well plates containing 500 µL of DMEM with 10% FBS for HEY and SKOV3, or 20% FBS for OV-90. After 24 h, the inserts were removed, and the upper side of the inserts was gently cleaned with swabs to remove non-migrated cells. Cells on the lower side of the inserts were then fixed with ice-cold methanol for 15 min and left to air dry. Crystal violet (Sigma-Aldrich; V5265) staining was performed by soaking the inserts in the dye for 10 min. The upper side of the inserts was again cleaned with swabs. Once the inserts were dry, three images per insert were acquired using an Olympus IX50 microscope equipped with an OPTIKA camera at 4× magnification. To quantify the migrated cells, crystal violet was solubilized by adding 200 µL of 33% acetic acid (Carlo Erba; 2789) to each insert. Then, 100 µL of the solubilized solution was transferred into a 96-well plate, and absorbance was measured at 595 nm using a Victor™ X4 plate reader (PerkinElmer).

### 2.11. Western Blot

Cells were seeded in 6-well plates at a density of 20,000 cells/cm^2^ and treated with 75% of the different CM in 1% FBS. After 24 h of treatment, the cells were collected and homogenized in RIPA buffer (supplemented with protease inhibitor, Sigma-Aldrich, P8340, and phosphatase inhibitor, Merck, P0044). Protein extraction was performed by applying 5 cycles of sonication, with cooling on ice between cycles. Protein concentration was determined using the BCA assay (Thermo Fisher Scientific; 23235). Next, 20 µg of protein per sample was loaded onto an SDS-PAGE gel and analyzed by Western blot using specific primary antibodies against phospho-p70 S6 Kinase (Cell Signaling Thecnologies, Danvers, MA, USA; 9205), and GAPDH (Bio-Rad; Hercules, CA, USA; MCA4740). After primary antibody incubation, membranes were probed with the appropriate secondary antibodies: anti-rabbit-HRP (Bio-Rad; 170-6515) or anti-mouse-HRP (Bio-Rad; 170-6516).

### 2.12. Spheroid Generation

Spheroids were generated by seeding 3000 cells for HEY or 6000 cells for SKOV3 and OV-90. To promote spheroid formation, 2% methylcellulose (Sigma Aldrich; M7027) was added to the respective culture medium, comprising 20% of the total volume. The cells were then seeded in U-bottom 96-well plates (Corning; 3788) in a final volume of 100 µL per well. After 24 h, some of the medium was replaced with different conditioned media (CM) to achieve final concentrations of 50% or 75% in 1% FBS. For the matrix assay, spheroids were treated for 6 days with increasing concentrations of CM (12.5%, 25%, 50%, 75%) in combination with increasing concentrations of paclitaxel (1.1, 3.3, 10, 30 nM). Alternatively, 75% CM was added immediately during spheroid formation under the same spheroid-forming conditions.

Spheroid Growth Assessment: To assess spheroid growth, 6–8 images per condition were captured using an Olympus IX50 microscope equipped with an OPTIKA camera at 4× magnification on day 1, day 3, and day 6. The areas of the spheroids were measured using ImageJ software (http://rsb.info.nih.gov/ij/) and reported in μm^2^.

ImageJ Calculations: Spheroid area was calculated using the “Macros” plugin in ImageJ. Images were first converted to grayscale. Images with an area less than 1000 pixels^2^ were excluded, and any holes within the spheroids were automatically filled. The area of the resulting spheroids was then calculated automatically in ImageJ software [24].

### 2.13. Three-Dimensional Cell Viability Assay (ATPlite)

Three spheroids per condition were harvested with 50 µL of medium after 6 days of treatment and collected into a 96-well plate. The assay was performed according to the manufacturer’s instructions (3D ATPlite 1 Step, PerkinElmer; 38221900). After adding 50 µL of lysis buffer to each well, the spheroids were mechanically disaggregated by pipetting 10–15 times. The plate was covered and incubated for 30 min on an orbital shaker in the dark. After incubation, 50 µL from each well was transferred into a 96-well white plate. Luminescence was measured using a Victor™ X4 luminometer (PerkinElmer).

### 2.14. Three-Dimensional Co-Culture Angiogenesis Model

The protocol was adapted from Xiao Wan et al. [25]. HUVEC were labeled with Celltracker green CMFDA 5 μM at 37 °C for 30 min. Then, 40 μL Geltrex was added into a 96-well plate which had been pre-chilled on ice. The plates were incubated at 37 °C for 30 min, allowing the Geltrex to polymerise. Then, 12,800 HUVECs resuspended in 40 μL EBM-2 2% FBS were seeded onto the polymerised gel layer.

After four hours of HUVEC seeding, 5000 ovarian cancer cells in 100 μL hAMSC-CM or hAM-CM with 2% FBS and 10% Geltrex (*v*/*v*), in the presence or absence of 16.6 nM Paclitaxel, were added onto the polymerized Geltrex. The final concentration of CM was 50%. The plate was then incubated at 37 °C to allow the top layer of Matrigel to polymerize. After 24 h, z-stack microphotographs with a step size of 4.75 µm were taken at Mica Widefield Live Cell (Leica Microsystems) using a 10× objective. Image reconstructions were performed using MAX projection of 15–20 images. Images were analyzed with the ImageJ Angiogenesis Analyzer Plugin (https://imagej.net/ij/macros/toolsets/Angiogenesis%20Analyzer.txt, accessed on 15 February 2025) [26]. Total segment length and the number of isolated segments were measured and calculated.

### 2.15. Statistical Analysis

Data report the mean and standard deviation. The parameters were compared using one-way or two-way analysis of variance (ANOVA), with Dunnet multiple comparison test post-analysis. N is reported in each figure legend. Statistical analysis was performed using Prism 9.5 (GraphPad Software, La Jolla, CA, USA). A *p*-value lower than 0.05 was considered statistically significant.

## 3. Results

### 3.1. hAMSC-CM and hAM-CM Inhibit Ovarian Cancer Cell Proliferation

We first characterized the antiproliferative effects of CM derived from hAMSC-CM and hAM-CM on three ovarian cancer cell lines: HEY, SKOV3, and OV-90. CM were collected after five days from pooled samples and assessed for their ability to reduce cell viability using the CyQUANT assay. HEY, SKOV3, and OV-90 cell lines were treated with different concentrations of CM (12.5%, 25%, 50%, 75%, and 100%) for 24, 48, 72, and 96 h. There was a dose-dependent inhibition of proliferation in all three cell lines, with effects becoming apparent after 48 h (Figure 1A,B). HEY cells were most sensitive to both hAMSC-CM and hAM-CM, while OV-90 cells exhibited the least sensitivity. Notably, CM from normal dermal fibroblasts had no effect on cell viability, underscoring the specificity of hAMSC-CM and hAM-CM (Appendix A). Mechanistically, hAM-CM markedly inhibited the phosphorylation of p70 S6 kinase (p-p70 S6), a key regulator of cell growth and cell cycle progression, after 24 h of treatment (Figure 1C).

Accordingly, in clonogenic assays, both CM types significantly reduced colony formation and size in all cell lines, with hAM-CM exhibiting the strongest effect (Figure 2A). To assess apoptosis, we analyzed treated cells via flow cytometry at 48 h. hAM-CM significantly induced apoptosis in HEY and OV-90 cells, while hAMSC-CM showed a less pronounced, non-significant effect. SKOV3 cells, lacking functional p53, showed no increase in apoptosis under either treatment (Figure 2B). These findings demonstrate that both hAMSC-CM and hAM-CM reduce cell viability in a dose-dependent manner, disrupt clonogenic potential, and induce apoptosis in a cell dependent manner.

### 3.2. hAMSC-CM and hAM-CM Differentially Affect 3D Spheroid Proliferation

Ovarian cancer metastasizes in part through spheroid formation and peritoneal dissemination. To explore the impact of CM on tumor cells, we assessed its effects on 3D aggregates, known as spheroids. The three tumor cell lines exhibited distinct spheroid morphologies. On the first day of formation, HEY cells formed compact, rounded spheroids, OV-90 cells formed rounded but less compact spheroids, and SKOV3 cells formed spheroids that were both less compact and less rounded. Spheroids were treated with 75% CM for six days, and their areas were measured on days 1, 3, and 6. The cell lines displayed differential responses to CM treatment. Both hAMSC-CM and hAM-CM significantly reduced the spheroid area of HEY and SKOV3 starting on day 3 (Figure 3A,B). Notably, SKOV3 spheroids exhibited a more compact morphology following CM treatment. In contrast, OV-90 spheroids treated with CM maintained the same area as the control group (Figure 3C).

ATP content, an indicator of cell viability [27], was measured in spheroids after six days of treatment with hAMSC-CM and hAM-CM. ATP levels were normalized to the control spheroids. Remarkably, in all cell lines, both hAMSC-CM and hAM-CM treatments significantly reduced ATP content (Figure 3D–F).

To further investigate, we evaluated spheroid formation and growth when hAMSC-CM and hAM-CM were added directly during spheroid generation. After one day of treatment, HEY and OV-90 spheroids appeared less aggregated in the presence of either CM, resulting in a larger apparent area compared to the control. By day 6, however, the spheroid area of HEY—and to a lesser extent, SKOV3 and OV-90—was reduced following CM treatment (Figure 4A–C). Additionally, ATP content in all cell lines was significantly reduced in the presence of CM, mirroring the results observed when CM was added after spheroid formation (Figure 4D–F).

In summary, these findings demonstrate that hAMSC-CM and hAM-CM decrease spheroid viability in a 3D model. This effect is observed in both pre-formed, treated with CM after 24 h of spheroid formation, and forming spheroids, where CM was added during the formation process, suggesting potential mechanisms of action on ovarian cancer cell aggregates.

### 3.3. hAMSC-CM and hAM-CM Inhibit Ovarian Cancer Cell Migration

CM derived from hAM has been reported to inhibit bladder cancer cell migration [11], while CM from other MSC sources, such as Wharton’s Jelly, has shown inhibitory effects on ovarian cancer cell migration [28]. Here, we examined how CM from hAMSC and hAM affects the migration of ovarian cancer cell lines. Using a wound-healing assay, we monitored the migratory ability of the cell lines in the presence of the two CM at 24 and 48 h. hAM-CM significantly inhibited the migration of all three cell lines in a dose-dependent manner. In contrast, hAMSC-CM reduced migration in HEY and OV-90 cells but had no effect on SKOV3 migration at 24 h. At 48 h, SKOV3 migration showed a slight, non-significant increase (Figure 5A–C). These findings were partially corroborated by single-cell migration assay (Figure 5D), where both CM reduced the total migration distance (path length) of HEY and OV-90 cells, while increasing SKOV3 migratory capacity. Interestingly, hAMSC-CM retained its inhibitory effects on HEY and OV-90 single-cell migration even after 24 h of pre-treatment, as shown in the transwell migration assay. Additionally, hAM-CM specifically inhibited HEY migration under these conditions (Appendix A). Collectively, these data suggest that hAMSC-CM and hAM-CM inhibit migration in both HEY and OV-90 cells, while having contrasting effects on SKOV3 migration, enhancing its single-cell and collective migration capacity.

### 3.4. hAMSC-CM and hAM-CM Affect Spheroid Invasion

In ovarian cancer, spheroids represent the minimal metastatic units capable of invading and implanting at distant sites. To investigate the impact of hAMSC-CM and hAM-CM on spheroid invasion, we evaluated the invasive ability of spheroids pre-treated with CM for 24 h within a surrounding matrix. HEY spheroids exhibited a strong invasive capacity, evident from 24 h onward. As shown in Figure 6A, spheroids pretreated with either hAMSC-CM or hAM-CM displayed a significantly reduced invasion area at both 24 and 48 h. SKOV3 spheroids, on the other hand, demonstrated a limited ability to invade the matrix, producing only a few protrusions from the spheroid core. Notably, after 48 h, treatment with either CM further inhibited their ability to form processes (Figure 6B). OV-90 spheroids also invaded the surrounding matrix, with a prominent invasion area observed on day 5. Interestingly, CM-pretreated OV-90 spheroids exhibited a smaller invasion area compared to the control group (Figure 6C). These findings suggest that both hAMSC-CM and hAM-CM effectively reduce the invasive capacity of ovarian cancer spheroids, impacting cell lines with varying invasive potentials.

### 3.5. Effect of hAMSC-CM and hAM-CM in Combination with Paclitaxel on Ovarian Cancer Cell Lines in 2D and 3D

To evaluate the potential enhanced therapeutic effect of hAMSC-CM and hAM-CM combined with paclitaxel in inhibiting ovarian cancer cell proliferation, combination treatments were conducted using both 2D and 3D models. These experiments included three ovarian cancer cell lines, with increasing concentrations of hAMSC-CM or hAM-CM and paclitaxel, to assess their combinatorial effects.

The combinatorial effects were analyzed using the ZIP reference model in SynergyFinder+ [29]. The results are shown in Appendix A for the 2D model and Figure 7 for the 3D model. Positive deviations between observed and expected responses, indicative of synergy, are represented in red, while negative deviations, indicative of antagonism, are shown in green. The relative dose–response matrices are showed in Appendix A, showing the percentage of cell viability relative to controls.

Both hAMSC-CM and hAM-CM demonstrated an enhanced therapeutic effect when combined with paclitaxel, particularly in the OV-90 cell line. This enhanced effect was observed consistently across both 2D and 3D models, underscoring the robust potential of these combinations to improve therapeutic outcomes. These findings suggest that hAMSC-CM and hAM-CM can amplify the efficacy of paclitaxel against ovarian cancer.

### 3.6. Effect of hAMSC-CM and hAM-CM in Combination with Paclitaxel in 3D Co-Culture Angiogenesis Model

Angiogenesis is a hallmark of cancer, and it is responsible for tumor spread and metastasis in ovarian cancer by promoting new blood vessel formation. It is essential for tumor growth and development. To assess the effect of hAMSC-CM and hAM-CM on tumor angiogenesis, we used a 3D co-culture angiogenesis model. We evaluated the tubule-like endothelial structures by HUVECs on top of a layer of Geltrex after 24 h of co-culturing with ovarian cancer cell lines. The analysis of digital images of tube-like networks was carried out using ImageJ angiogenesis analyzer software. We measured the total segment length and number of isolated segments, two parameters with opposite trends when angiogenesis is inhibited. Specifically, a reduction in total segment length and an increase in the number of isolated segments indicate impaired angiogenesis. As shown in Figure 8, hAMSC-CM had no impact on angiognesesis, on the contrary, for all three ovarian cancer cell lines, hAM-CM impaired the formation of tube structures. Interestingly, angiogenesis was severely impaired when both hAMSC-CM and hAM-CM were combined with paclitaxel.

## 4. Discussion

Ovarian cancer remains a formidable challenge in oncology, with high mortality rates due to late diagnosis, tumor recurrence, and resistance to conventional therapies. Despite advances in surgical techniques and the development of chemotherapeutics such as paclitaxel, the clinical outcomes for ovarian cancer patients remain suboptimal. This underscores a critical unmet need for innovative therapeutic approaches that target the biological underpinnings of tumor progression and metastatic dissemination. In this study, we explored the therapeutic potential of CM derived from hAMSC and intact hAM on ovarian cancer cell lines in both 2D and 3D models. These secretomes, enriched in bioactive molecules, offer a cell-free and scalable therapeutic strategy that could complement existing treatments. Our findings highlight the ability of these CM to inhibit multiple hallmarks of cancer progression, including cancer cell proliferation, migration, invasion, angiogenesis, and spheroid growth. Our results demonstrate that both hAMSC-CM and hAM-CM effectively inhibit ovarian cancer cell proliferation in a dose-dependent manner (Figure 1). Moreover, the combination of CM with paclitaxel enhanced the anti-tumor effects on proliferation in 2D cultures and spheroid growth.

Notably, hAM-CM consistently displayed stronger antiproliferative and pro-apoptotic effects compared to hAMSC-CM, likely reflecting the combined contributions of bioactive factors secreted by both stromal and epithelial cells within the amniotic membrane.

The differential response observed among the ovarian cancer cell lines—HEY, SKOV3, and OV-90—emphasizes the importance of tumor heterogeneity. HEY cells, which harbor wild-type p53, were the most sensitive to CM treatment, while OV-90 cells, characterized by a missense TP53 mutation, exhibited the least sensitivity. These findings are consistent with the notion that p53 status influences cellular responses to stress and therapeutic interventions [30,31]. Previous studies have reported similar antiproliferative effects of MSC-derived CM in various cancer types, attributed to the secretion of bioactive molecules, including cytokines, growth factors, and extracellular vesicles [10,32,33,34,35]. The stronger effects observed with hAM-CM may reflect the combined contributions of hAMSCs and hAECs, which are known to possess complementary anticancer properties [14,36,37,38,39,40].

In addition to reducing cell proliferation, hAM-CM significantly induced apoptosis in HEY and OV-90 cells, as evidenced by increased Annexin V/PI staining (Figure 2). In contrast, SKOV3 cells, which are p53-null, showed minimal apoptotic responses, further supporting the hypothesis that CM-induced apoptosis may depend, at least in part, on p53-mediated pathways. This mechanistic link between p53 status and CM efficacy highlights the potential for tailoring secretome-based therapies to the molecular characteristics of specific tumor subtypes [41].

Spheroids represent a more physiologically relevant model of tumor biology, closely mimicking the architecture and microenvironment of in vivo tumors [42]. Our study revealed that both hAMSC-CM and hAM-CM reduced spheroid growth and ATP content, indicating impaired viability (Figure 3). Interestingly, CM disrupted spheroid formation when introduced during aggregation (Figure 4), suggesting potential interference with cellular adhesion and extracellular matrix interactions [43,44]. These findings are particularly significant given the role of spheroids in ovarian cancer metastasis and chemoresistance [45]. The observed effects highlight the potential of hAMSC-CM and hAM-CM to target both established and forming tumor microenvironments.

Cancer metastasis is a major cause of mortality in ovarian cancer, with migration and invasion being critical steps in this process [46,47]. Both hAMSC-CM and hAM-CM inhibited the migratory capacity of HEY and OV-90 cells, as demonstrated in wound-healing, transwell migration, and single-cell motility assays (Figure 5). However, SKOV3 cells exhibited a unique response, with increased migration observed under certain conditions. This divergence may reflect differences in intrinsic cell motility mechanisms or interactions with CM components [11,48]. Additionally, the reduction in spheroid invasion (Figure 6) into the matrix after CM pretreatment underscores the potential of these secretomes to disrupt metastatic progression [49].

To further explore the therapeutic potential of CM, we investigated its combination with paclitaxel, a first-line chemotherapeutic agent in ovarian cancer treatment. The combination treatment demonstrated enhanced therapeutic effects in both 2D and 3D models (Appendix A and Figure 7), particularly in the OV-90 cell line. Using the ZIP reference model implemented in SynergyFinder+, we identified a significant interaction between CM and paclitaxel, with positive deviations indicative of synergy. This enhanced therapeutic effect may result from the complementary actions of paclitaxel, which disrupts microtubule dynamics and induces mitotic arrest [50,51], and the bioactive components of CM, which modulate key signaling pathways implicated in tumor survival, proliferation, and metastasis. Previous studies have linked CM-derived molecules to the inhibition of IL-6/JAK2/STAT3 and FAK/PI3K/Akt/mTOR pathways, providing potential mechanistic insights into the observed enhanced effect [9,11,52,53]. Paclitaxel has been reported to induce the upregulation of S6 as a compensatory adaptive response [54], and previous studies have shown that targeting p70 with specific inhibitors can enhance Paclitaxel efficacy in reducing cancer cell viability. While our interpretation remains speculative, the observed reduction in p-p70 S6 in our model may suggest a similar mechanism, potentially contributing to the enhanced effect of Paclitaxel. Notably, hAMSC can be loaded with paclitaxel and the drug released from hAMSC is able to inhibit pancreatic cancer cell line proliferation in vitro [55]. Interestingly, not only does the released paclitaxel impact proliferation but also hAMSC are able to inhibit tumor cell proliferation per se under specific culture conditions in vitro.

The mechanisms underlying the anticancer effects of hAMSC-CM and hAM-CM are likely multifactorial, involving the secretion of bioactive molecules that modulate cell signaling, immune responses, and the tumor microenvironment [19,56,57]. While our study establishes the therapeutic potential of these CM, further research is needed to elucidate the specific components responsible for their effects. Proteomic and transcriptomic analyses of CM could provide valuable insights into its active constituents and mechanisms of action. Additionally, in vivo studies are essential to validate the efficacy and safety of CM-based therapies and to assess their potential for clinical translation. Additional experiments are required to gain a deeper understanding of the mechanisms by which the amniotic membrane influences cancer dynamics, and to clarify its potential as an adjuvant therapeutic strategy for targeting tumor cells. Interestingly, we observed that hAM-CM alone is able to affect capillary-like structure formation, while hAMSC-CM had no effect (Figure 8). The combination with paclitaxel enhanced the effect, suggesting that the combinatorial approach, especially for hAM-CM, could have an impact also on the tumor microenvironment, which plays a pivotal role in tumor growth and dissemination. These findings prompt us to further investigate this aspect in different experimental models to better elucidate the impact of CM on tumor angiogenesis. Given the dual role of tumor vasculature in both sustaining tumor progression and modulating drug delivery, it will be crucial to assess not only the inhibition of angiogenesis but also the potential ability of CM to normalize the vasculature.

In conclusion, this study demonstrates that hAMSC-CM and hAM-CM exhibit potent antitumor activity against ovarian cancer by targeting critical processes in tumor progression, including proliferation, migration, invasion, and spheroid growth. The consistent enhanced effect observed across multiple experimental conditions highlights the potential of CM as an adjuvant therapy to improve clinical outcomes while potentially reducing the toxicities associated with high-dose chemotherapy. These findings position the CM as a promising, cell-free therapeutic strategy in the fight against ovarian cancer, warranting further investigation in preclinical and clinical settings.

## Figures and Tables

**Figure 1 pharmaceutics-17-00420-f001:**
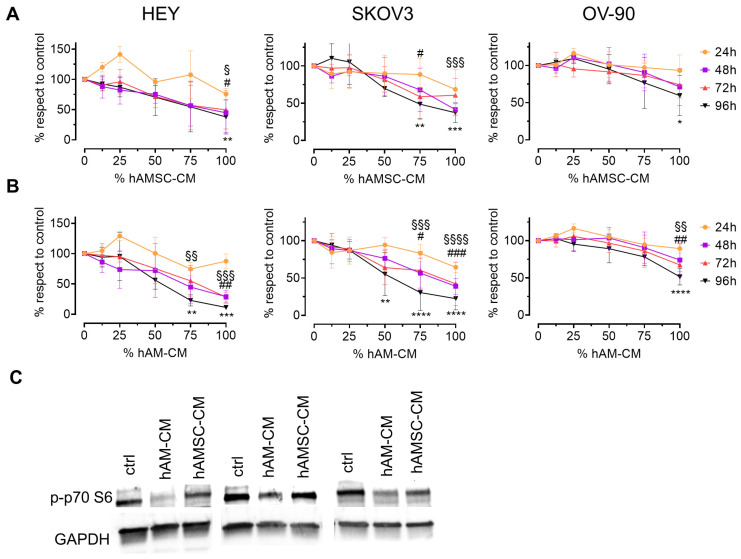
hAMSC-CM and hAM-CM inhibit ovarian cancer cell proliferation. HEY, SKOV3, and OV-90 cell lines were treated with differing concentrations of hAMSC-CM (**A**) or hAM-CM (**B**) (12.5%, 25%, 50%, 75%, and 100%) for 24, 48, 72, and 96 h. Cell viability was assessed using the CyQUANT assay, *n* = 3–5. Results are expressed as a percentage relative to untreated controls at each time point (orange: 24 h; purple: 48 h; red: 72 h; black: 96 h). Data are presented as mean ± SD, with significance levels indicated (for 48 h: ^§^ *p* < 0.05, ^§§^ *p* < 0.01, ^§§§^ *p* < 0.001, ^§§§§^ *p* < 0.0001; for 72 h: # *p* < 0.05, ## *p* < 0.01, ### *p* < 0.001; for 96 h: * *p* < 0.05, ** *p* < 0.01, *** *p* < 0.001, **** *p* < 0.0001). (**C**) Western blot for p-p70 S6 were performed on protein lysates of HEY, SKOV3 and OV-90 treated with 75% CM for 24 h.

**Figure 2 pharmaceutics-17-00420-f002:**
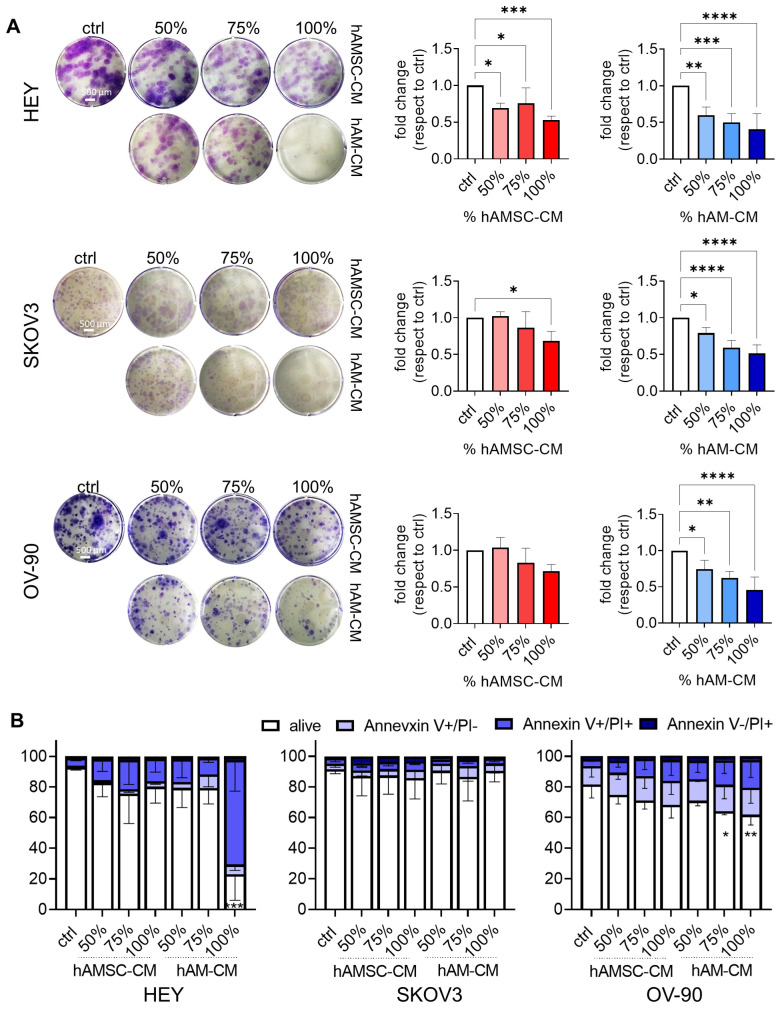
hAMSC-CM and hAM-CM affect ovarian cancer cell colony formation and apoptosis. (**A**) Colony formation assays for HEY, SKOV3, and OV-90 cell lines treated with 50%, 75%, and 100% CM. Representative images of stained colonies are shown (**left** panel, scale bar 500 µm), with quantification of colony formation expressed as fold change relative to controls (**right** panel, red: hAMSC-CM, blue: hAM-CM), *n* = 3–4. (**B**) Apoptosis analysis of HEY, SKOV3, and OV-90 cell lines treated with 50%, 75%, and 100% CM for 48 h. Apoptosis was detected by Annexin V/PI staining and quantified using flow cytometry. Results are shown as percentages of apoptotic cells. *n* = 3–4. Data are presented as mean ± SD, with significance levels indicated (* *p* < 0.05, ** *p* < 0.01, *** *p* < 0.001, **** *p* < 0.0001).

**Figure 3 pharmaceutics-17-00420-f003:**
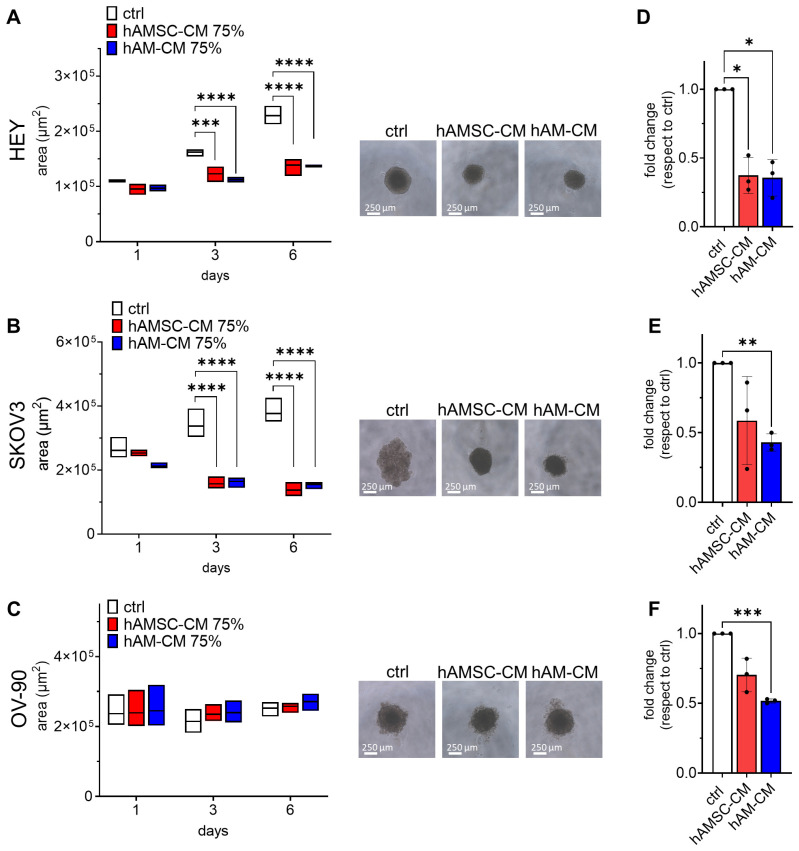
hAMSC-CM and hAM-CM reduce ovarian cancer spheroid growth and ATP content. Spheroids from HEY (**A**), SKOV3 (**B**), and OV-90 (**C**) cell lines were treated with 75% CM for six days. Spheroid areas were measured at days 1, 3, and 6 using ImageJ and expressed in μm^2^ (white: control; red: hAMSC-CM; blue: hAM-CM). Representative micrographs of spheroids at day 6 are shown (right panel, scale bar 250 μm). (**D**–**F**) The ATP content of HEY, SKOV3, and OV-90 spheroids was measured after six days of treatment and expressed as fold change relative to untreated controls. Data are presented as mean ± SD. *n* = 3. Significance levels are indicated (* *p* < 0.05, ** *p* < 0.01, *** *p* < 0.001, **** *p* < 0.0001).

**Figure 4 pharmaceutics-17-00420-f004:**
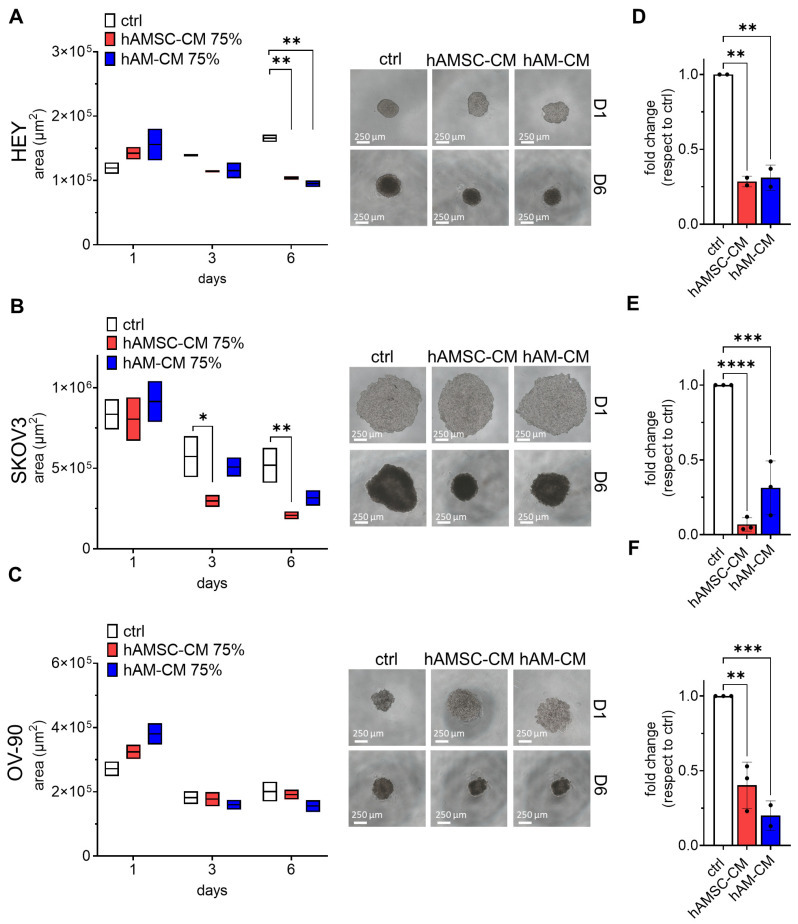
CM pretreatment affects spheroid growth and ATP content. Spheroids from HEY (**A**), SKOV3 (**B**), and OV-90 (**C**) cell lines were treated with 75% CM during formation and monitored for six days. Spheroid areas were measured at days 1, 3, and 6 using ImageJ and expressed in μm^2^ (white: control; red: hAMSC-CM; blue: hAM-CM). Representative micrographs of spheroids at day 1 and day 6 are shown (right panel, scale bar 250 µm). (**D**–**F**) ATP content of HEY, SKOV3, and OV-90 spheroids was measured after six days of CM pretreatment and expressed as fold change relative to untreated controls. *n* = 3. Data are presented as mean ± SD, with significance levels indicated (* *p* < 0.05, ** *p* < 0.01, *** *p* < 0.001, **** *p* < 0.0001).

**Figure 5 pharmaceutics-17-00420-f005:**
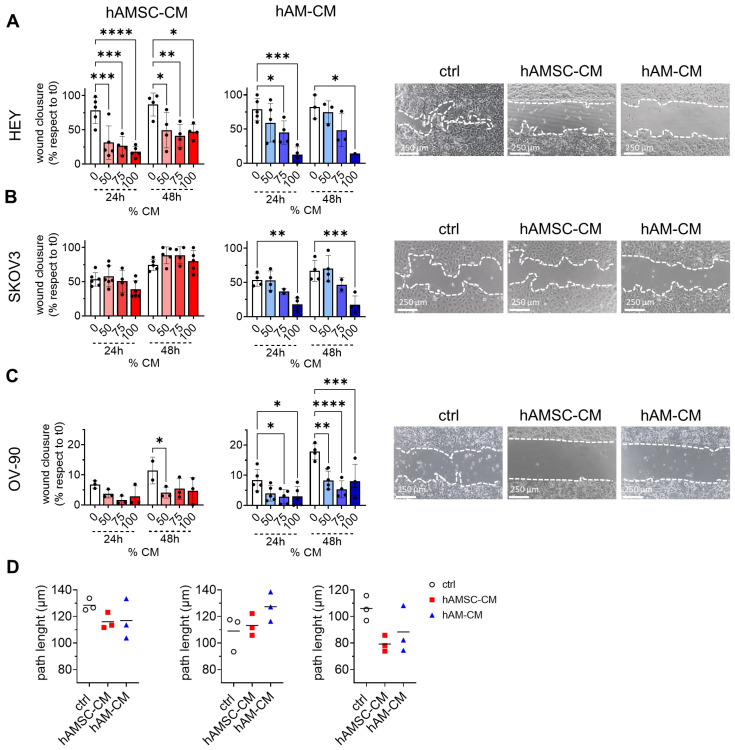
hAMSC-CM and hAM-CM inhibit ovarian cancer cell migration. Wound-healing assays were performed on monolayers of HEY (**A**), SKOV3 (**B**), and OV-90 (**C**) cells treated with 50%, 75%, and 100% CM. Wound closure was measured at 0, 24, and 48 h and expressed as the percentage of the initial wound area (white: control; red: hAMSC-CM; blue: hAM-CM). Representative micrographs of the wound area at 75% CM treatment are shown (right panel, scale bar 250 µm), with white dotted lines marking the wound boundaries. *n* = 3–5. Data are presented as mean ± SD (* *p* < 0.05, ** *p* < 0.01, *** *p* < 0.001, **** *p* < 0.0001). (**D**) Single-cell migration was performed on HEY, SKOV3, and OV-90 cells treated with 75% CM by time-lapse analysis. Each point is a replicate and refers to the mean of at least 30 cell paths for each condition.

**Figure 6 pharmaceutics-17-00420-f006:**
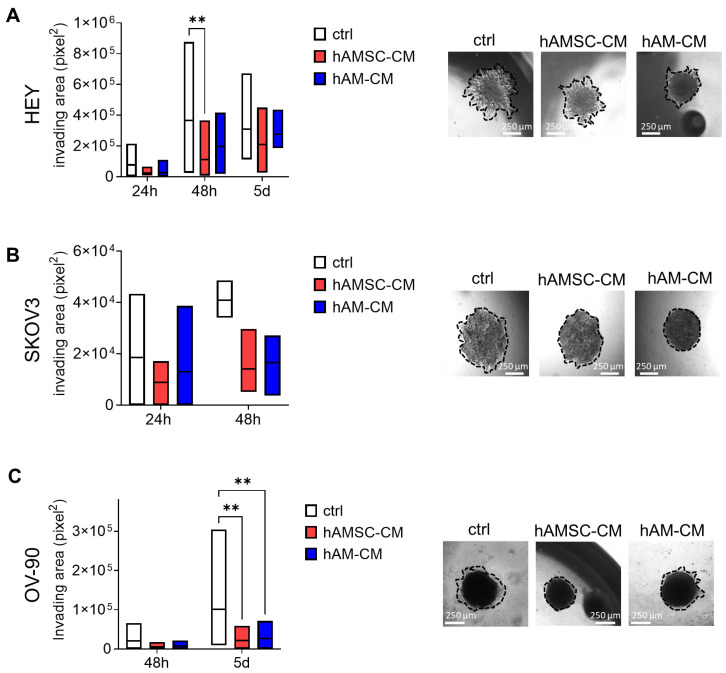
hAMSC-CM and hAM-CM inhibit spheroid invasion. Spheroids from HEY (**A**), SKOV3 (**B**), and OV-90 (**C**) cells were pretreated with 75% CM for 24 h and embedded in a Geltrex matrix. Invasion areas were measured after 24, 48, and 120 h and expressed in μm^2^ (white: control; red: hAMSC-CM; blue: hAM-CM). Representative micrographs of invading spheroids are shown (right panel, scale bar 250 μm), with invasion boundaries marked by black dotted lines. Data are presented as mean ± SD (** *p* < 0.01).

**Figure 7 pharmaceutics-17-00420-f007:**
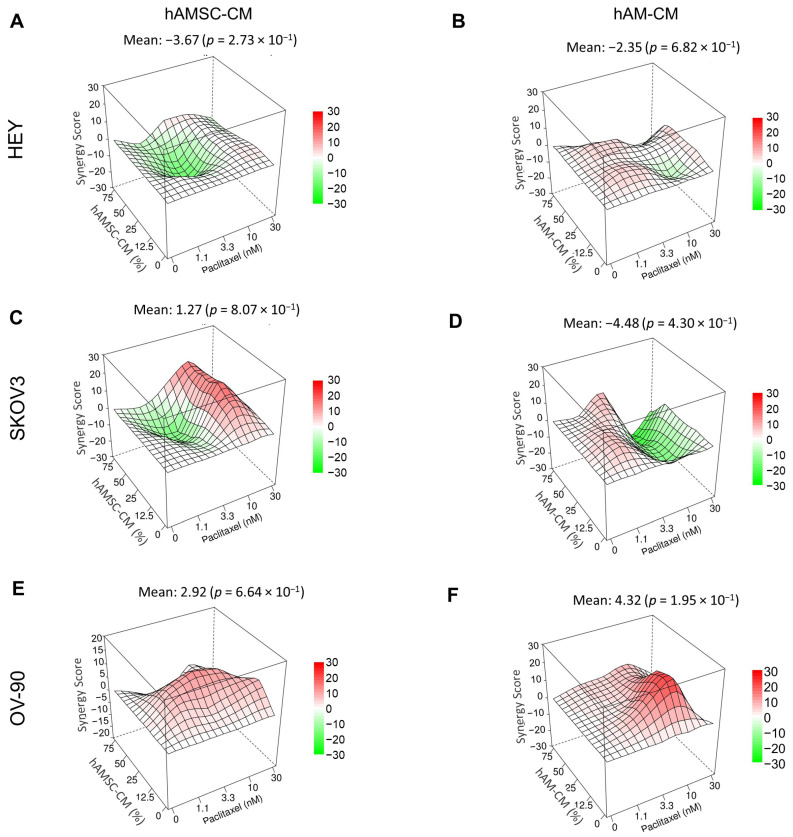
Synergistic effects of CM and paclitaxel on ovarian cancer spheroids in 3D. Spheroids from HEY (**A**,**B**), SKOV3 (**C**,**D**), and OV-90 (**E**,**F**) cells were treated with different concentrations of CM (12.5%, 25%, 50%, 75%) and paclitaxel (1.1, 3.3, 10, 30 nM) for 48 h. Viability was assessed using the ATP-lite assay. Results are displayed as ZIP synergy maps and synergy scores for each combination and cell line are shown.

**Figure 8 pharmaceutics-17-00420-f008:**
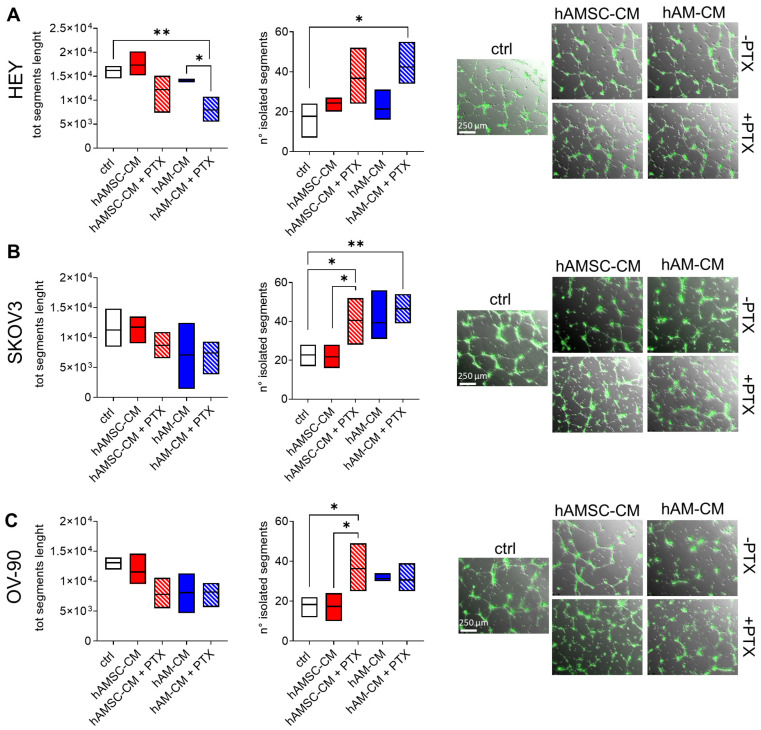
Effects of CM and paclitaxel in a co-culture angiogenesis model. Co-culture of HUVECs and ovarian cancer cell lines in a Geltrex sandwich. HUVEC were seeded on a Geltrex later for 4 h, then HEY (**A**), SKOV3 (**B**), OV-90 (**C**) with 50% hAMSC-CM or hAM-CM in the presence or absence of 10 nM paclitaxel (PTX) were seeded on the top. Tubule-like endothelial structures were analyzed after 24 h with the ImageJ Angiogenesis analyzer plugin. Data are expressed as total segment length and number of isolated segments. N = 3 (* *p* < 0.05; ** *p* < 0.01). On the right, representative micrographs of different treatments. Scale bar 250 µm.

## Data Availability

The original contributions presented in this study are included in the article/Appendix A. Further inquiries can be directed to the corresponding author.

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
