# Peer review of "Synergistic Effect of Conditioned Medium from Amniotic Membrane Mesenchymal Stromal Cells Combined with Paclitaxel on Ovarian Cancer Cell Viability and Migration in 2D and 3D In Vitro Models"

_pharmaceutics, 2025, doi:10.3390/pharmaceutics17040420_

Round 1
Reviewer 1 Report
Comments and Suggestions for Authors
Dear colleagues!
After reviewing the manuscript by Chiodelli et al. I have the following comments regarding the study and manuscript. The work covers and observational study on impact of conditioned media from human amniotic membrane MSC on growth, migration and invasion of ovarian cancer cell lines in vitro as 2D or 3D (spheroid) cultures. One would praise use of 3 reference cell lines of human ovarian cancer and pinpointing heterogeneous response of the latter in Discussion as well as overall diligence during experiments using N=3 for donor power of the study. Methods are basically routine cell biology assays provided in sufficient replicates and original WBs are provided along with manuscript's files.
However, certain points might be improved to provide the Reader a more interesting communication:
1) it is unclear what served as a negative control in most of experiments. Generally, it should be provided as "empty" CM (e.g. 1% FBS DMEM/F12 which was used to make up CMs from MSC) freshly added to control wells in 0h point. Frankly, the assay should mock the medium change which occurred in CM samples to account for putative impact of manipulation and addition of fresh medium. Furthermore, if medium was not replaced in control samples starvation and turnover of FBS components might impact the cells. Thus, it should be clearly explained and provided in M&Ms section
2) Some figures are overburdened by panels which makes them really hard to read. Mostly this point relates to Fig.1 which should be definitely split in 2 - proliferation and colony/survival figures - bars and plots are hard to ready in such resolution as provided. Please, have a closer look at others to ensure understandable size and resolution of images.
3) tumor vascularization is a really important point for progression and thus one would definitely be interested in results of any angiogenesis assay (e.g., HUVEC or Ea926hy co-culture) in this work
4) finally, the study' pivotal flaw (in terms of physiological mechanism understanding) is observational nature of the work - little is provided regarding mechanisms behind observed effects. In light of this comment it is really unclear why would Authors move WB images to supplementary when this is the only target-related finding in the work. In case p70 is known to be impacted by Paclitaxel as well it provides at least a speculative point for Discussion.
5) For 2-page long discussion references to Figures that support claims and conclusions would be a great aide to a potential Reader.
Regards, Reviewer
Reviewer 2 Report
Comments and Suggestions for Authors
The paper is an interesting in vitro study on the antitumor effects exerted by the secretome of the mesenchymal stromal cells from the human amniotic membrane (hAMSC) and the intact amniotic membrane (hAM). The authors raised the experimental design to 3D models, in order to obtain more accurate, translatable results for potential therapeutic use in aggressive, resistant ovarian cancer. The paper is suitable for the journal, easy to read by a wide audience, and the conclusions are based on reproducible experimental data, evaluated with an open-access analysis tool.
I strongly recommend publishing the study; some minor revisions can be done, such as:
Page 3, chapter 2.2 Human amniotic membrane (hAM) fragment preparation- authors should add data on ethical approvals from the host institution/ surveillance authority for placenta usage, or to state if cells were harvested following the written informed consent.
Page 3, Ch 2.2 “cut into a 50 cm2 fragments” please revise the formulation and please specify if all fragments were of 50 square centimeters. Authors wrote later, in chapter 2.4 Preparation of conditioned medium (CM): “From hAM: Two fragments of hAM were placed in 50-mL conical tubes” two 50cm2 fragments in one 50ml tube? What was the volume of cell culture media in the tubes?
Page 4, row 158: please revise in the apoptosis method description: “the cells were resuspended in buffe and supplemented with Annexin V-FITC and PI. The cells were incubated (……). Following this, PI was added….”
Page 5, chapter 2.9- More details regarding the assessment/recording of cell migration paths and parameters should be briefly described. Later in the manuscript the authors do not mention the motility assay with the same name, it is somewhat difficult to find exact correspondence between method 2.9 and the result. Details are needed for a better understanding of Figure 4D and the results description on page 12.
Page 5, chapter 2.12- the spheroid generation was made under the same conditions and same incubator as described earlier?
Page 7 “TP53-dependent contexts” “emerges as a potent paracrine agent” “selectively” please insert the appropriate evidences (TP53 measurement, measurements on paracrine regulators). In the Discussion section, on page 15, all those aspects are well explained, based on literature data and TP-mutations of the different cell lines, therefore this phrase could be more suitable for the Discussion chapter.
Page 8, chapter 3.2 “Spheroids were treated with 75% CM (..)” and “This effect is observed in both pre-formed and forming spheroids” – please specify the moments in the spheroid development (eg. day 1, day 3, or other..) when they were subjected to the 75% CM treatment.
Reviewer 3 Report
Comments and Suggestions for Authors
The paper contains many valuable results. I only have a few comments.
The last sentence of the introduction “Our results demonstrate that hAMSC-CM and hAM-CM significantly inhibit the proliferation and … demonstrate synergistic effects when the CM is combined with paclitaxel. “belongs to the conclusions and not to the introduction. Here a hypothesis should be formulated which will be proved in the experiments.
For instance, that you expect essential differences in the results obtained by 2D and 3D cell culture experiments.
Please remove conclusions from the paragraph results. For example: “These findings suggest that hAMSC-CM and hAM-CM can amplify the efficacy of paclitaxel against ovarian cancer, offering a promising strategy to overcome limitations of standard therapies. Further mechanistic studies are needed to elucidate the pathways contributing to this enhanced therapeutic effect and to explore its translational potential.” It belongs to the discussion.
Please check carefully the references and give also doi.
Reference 17 is a group from Argentina and I assume they do not belong to the authors.
I miss the Ethics Committee's vote on the use of human materials. We are not allowed to use human material for research purposes without the approval of the Ethics Committee, even if it is waste.
There are some typing errors. Please correct.
Materials and Methods
The seller data is incomplete.
Please give additional information to SKOV3 cells in line 77 and not in line 90. The same for the OV-90 cells. Did Daniela Gallo isolate these cells? Then an ethical votum is necessary.
Line 106 please give the value and not low g
MTT: Please include a hint that you measured the metabolic activity of your cells, and you considered this as “viability” without counting living and dead cells.
How many donor-CM were pooled (day 5)? Maybe I missed it.
Results and discussion are fine for me.
